# Antiviral Activities of Andrographolide and Its Derivatives: Mechanism of Action and Delivery System

**DOI:** 10.3390/ph14111102

**Published:** 2021-10-28

**Authors:** Sya’ban Putra Adiguna, Jonathan Ardhianto Panggabean, Akhirta Atikana, Febriana Untari, Fauzia Izzati, Asep Bayu, A’liyatur Rosyidah, Siti Irma Rahmawati, Masteria Yunovilsa Putra

**Affiliations:** 1Department of Chemistry, Faculty of Mathematics and Natural Sciences, Universitas Gadjah Mada, Bulaksumur, Yogyakarta 55281, Indonesia; syaban.putra@mail.ugm.ac.id (S.P.A.); jonathanpanggabean@mail.ugm.ac.id (J.A.P.); 2Research Center for Biotechnology, Research Organization for Life Sciences, National Research and Innovation Agency (BRIN), Jalan Raya Jakarta-Bogor KM. 46, Cibinong 16911, Indonesia; akhirta.atikana@gmail.com (A.A.); untariuntari72@gmail.com (F.U.); fauzia.izzati@gmail.com (F.I.); asep044@brin.go.id (A.B.); 3Research Center for Biology, Research Organization for Life Sciences, National Research and Innovation Agency (BRIN), Jalan Raya Jakarta-Bogor KM. 46, Cibinong 16911, Indonesia

**Keywords:** *Andrographis paniculata*, andrographolide, antivirals, mechanism of action, drug delivery system

## Abstract

*Andrographis**paniculata* (Burm.f.) Nees has been used as a traditional medicine in Asian countries, especially China, India, Vietnam, Malaysia, and Indonesia. This herbaceous plant extract contains active compounds with multiple biological activities against various diseases, including the flu, colds, fever, diabetes, hypertension, and cancer. Several isolated compounds from *A. paniculata*, such as andrographolide and its analogs, have attracted much interest for their potential treatment against several virus infections, including SARS-CoV-2. The mechanisms of action in inhibiting viral infections can be categorized into several types, including regulating the viral entry stage, gene replication, and the formation of mature functional proteins. The efficacy of andrographolide as an antiviral candidate was further investigated since the phytoconstituents of *A. paniculata* exhibit various physicochemical characteristics, including low solubility and low bioavailability. A discussion on the delivery systems of these active compounds could accelerate their development for commercial applications as antiviral drugs. This study critically reviewed the current antiviral development based on andrographolide and its derivative compounds, especially on their mechanism of action as antiviral drugs and drug delivery systems.

## 1. Introduction

*Andrographis paniculata* (Burm.f.) Nees is a herbaceous plant of the family Acanthaceae and is commonly known as “Chiretta” [1], “King of Bitters” or “Kalmegh” [2]. It has been used as a traditional medicine in Asian countries, such as China, India, Vietnam, Malaysia, and Indonesia. It is regarded as a safe and non-toxic medicine in traditional Chinese medicine (TCM) [3]. Usually, the aerial parts, roots, or leaves of *A. paniculata* are used as powders, infusions, or decoctions. *A. paniculata* has an excellent potency among medicinal plants due to its broad spectrum of biological activities related to the treatment of infectious and degenerative diseases. Clinical studies of standardized commercial *A. paniculata* extracts were shown to have excellent efficacy. For instance, KalmCold^TM^ was clinically shown to have an efficacy that is 2.1 times higher than that of a placebo (*p*-value ≤ 0.05) at reducing symptoms of upper respiratory tract infections in 223 patients [1]. A placebo-controlled double-blind study of Kan Jang^TM^ showed better efficacy of *A. paniculata* extracts than a placebo for reducing clinical symptoms among 59 common cold patients [4]. This evidence indicates that *A. paniculata* has a high market value, either in the form of its parts or standard compounds (e.g., 5 USD kg^−1^ and 100,000 USD kg^−1^ for good-quality dried leaves and andrographolide, respectively) [5].

The biological activities of *A. paniculata* could be associated with the presence of approximately 142 secondary metabolites in this plant’s tissues, including entalabdane diterpenoids, flavonoids, quinic acid derivatives, xanthones, rare noriridoids, steroids, and other compounds [5,6]. Most of them are extracted from the aerial parts, leaves, and whole parts of the plant, while some are obtained from the roots (Figure 1). Among the identified compounds, andrographolide and its derivatives, including neoandrographolide, 14-deoxy-11,12-dehydroandrographolide, and isoandrographolide, contribute the main reported pharmacological activities [2]. Andrographolide is used in many fields of medicine, including as an anticancer [7], antimicrobial [8,9], antioxidant [10], anti-inflammatory [11], antiviral [3,12,13], antidiabetic [14] and wound-healing agent [15]. Meanwhile, 14-deoxy-11,12-didehydroandrographolide exhibits antifungal, antiviral, and anticancer activities. Moreover, isoandrogrpaholide suppressed tumors and inhibited pro-inflammatory mediators (NO, IL-1β, and IL-6) and allergic mediators (LTB4 and histamine) [16,17]. Furthermore, neoandrographolide acts as an antiviral, anti-inflammatory, hepatoprotective, and antiradical agent [18,19].

Nowadays, the coronavirus disease 2019 (COVID-19) pandemic, caused by a severe acute respiratory syndrome coronavirus 2 (SARS-CoV-2) infection, has transformed the philosophy of drug discovery into “*one drug, multi-target*” instead of “*one drug, one target*” [2]. Due to COVID-19, much attention has been given to active biological compounds with potential antiviral activities. Many of them (both synthetic and natural substances) are being explored to gain in-depth information on their potential action against SARS-CoV-2. However, repurposing natural phytomolecules in medicinal plants has been regarded as an ideal (and the most economical) method of finding drug components of interest as quickly as possible [20]. 

*Andrographis paniculata* extract and some of its active compounds were reported to possess antiviral activities. For instance, andrographolide protects against deoxyribonucleic acid (DNA) viruses (e.g., Epstein–Barr (EB) virus and herpes simplex virus 1 (HSV-1)) and ribonucleic acid viruses (e.g., human immunodeficiency virus (HIV), influenza virus A, hepatitis C, dengue virus, and Japanese encephalitis virus (JEV)) [21]. An in silico study showed that andrographolide possessed potential inhibitors of SARS-CoV-2’s main protease [17,19,20]. In addition, 14-deoxy-11,12-dehydroandrographolide was effective in inhibiting the influenza A virus, HSV-1, and HIV replication. Moreover, 14-deoxyandrographolide exhibits antiviral activity to protect against human papillomavirus (HPV), HIV, and HSV-1 [22].

The efficacy of andrographolide (and its analogs) as an antiviral candidate requires further investigation. Although several reports and reviews have discussed their antiviral properties [2,3,23], the information on their mechanism of action and the understanding of their physicochemical characteristics are still limited. For example, despite its potential antiviral activity, the use of andrographolide is limited due to its low solubility in aqueous extracts, bitter taste, low stability in the gastrointestinal tract, and low bioavailability [24,25,26]. Thus, the dosage form of andrographolide and its derived compounds should be carefully considered when designing treatments.

Investigating the drug delivery systems of these phytoconstituents of *A. paniculata* has become a popular research topic recently. Several papers have reviewed the antiviral activity of andrographolide and its derivatives (see, for example, the work of Gupta et al. [21], Jiang et al [3], and Reshi and Yong [27]. However, discussing the antiviral activities of andrographolide and its derivatives, combined with the delivery system of andrographolide, is a more interesting topic. This is because it provides insights into how the pharmacological activities of andrographolide and its derivatives can be optimized when applied to biological systems.

Therefore, this review discusses the current state of using these two aspects of andrographolide and its analog compounds to develop antiviral drugs. The main purpose of this review was to present correlations between the physicochemical characteristics of these phytochemicals of *A. paniculata* with their antiviral activity and delivery systems as drugs. In this way, this review is expected to provide insight into how to accelerate the development of an antiviral drug based on andrographolide and/or its analog compounds

## 2. Literature Search Strategy

A systematic search was conducted through several databases (PubMed, Elsevier, American Chemical Society, Google Scholar, MDPI, Springer, and SciFinder) to compile articles published from 2010 to 2021 that contained terms such as “andrographolide and its derivatives”, “*Andrographis paniculata*”, “antiviral activities”, “mechanism of action”, and “delivery systems” in their titles and abstracts.

The searches generated 52 articles containing reports of the effects of andrographolide and its derivatives against influenza, herpes simplex, hepatitis, HIV, and SARS-CoV-2, as well as their mechanisms of action. Another 24 articles referring to andrographolide delivery systems were also found. Further searches for information about antivirus activity, mechanism of action, and delivery system were carried out with more specific keywords using the same sources.

## 3. Phytomolecules of Andrographis Paniculate

### 3.1. Andrographolide

As the main component of *A. paniculata*, andrographolide is an ent-labdane diterpenoid with a chemical formula of C_20_H_30_O_5_ (Figure 2) and a molecular weight of 350.45 g/mol. Andrographolide is a colorless crystalline solid plate with a melting point of 205–210 °C. The ultraviolet spectra in methanol and ethanol ranged from 223 to 224 nm (λ_max_) [28,29]. Andrographolide is easily dissolved in methanol, ethanol, pyridine, acetic acid, and acetone but only slightly dissolves in ether and water. Although it can be isolated using water-based solvents, it has a low solubility (46 mg L^−1^) in water, resulting in low bioavailability in living systems [30,31].

### 3.2. Neoandrographolide

Neoandrographolide can be found in the whole plant. It is isolated by methanol, ethanol, and acetone solvents. Neoandrographolide has a chemical formula of C_26_H_40_O_8_ (Figure 2) and a molecular weight of 480.59 g/mol. It forms a long colorless needle crystal and has a melting point of 167–168 °C. Specific optical rotations are −48° (pyridine) and −45° (absolute methanol). The UV spectrum showed that the maximum lambda (λ_max_) is 240 nm in a methanol solvent [32]. Neoandrographolide is soluble in methanol, ethanol, acetone, and pyridine but only slightly soluble in chloroform and water. In addition, it is insoluble in ether and petroleum ether [33,34].

### 3.3. 14-Deoxyandrographolide

Another major compound of *A. paniculata* is 14-deoxyandrographolide. 14-Deoxyandrographolide has been isolated from leaves, the aerial part, and the whole plant of *A. paniculata* with methanol, ethanol, ethyl acetate, and ethanol-water. It forms a colorless cylindrical crystal with a melting point of 170 °C. 14-Deoxyandrographolide showed a UV spectrum at 238 nm (λ_max_ in methanol). Its chemical formula is C_20_H_30_O_4_ (Figure 2) and its molecular weight is 334.44 g/mol [35,36,37]. 

### 3.4. Isoandrographolide

Isoandrographolide is a minor diterpenoid lactone that is isolated from *A. paniculata*. Isoandrographolide has the same chemical formula as andrographolide (C_20_H_30_O_5_) but it has a different geometry (Figure 2). Isoandrographolide forms a colorless prism with a melting point of 198–200 °C. Isoandrographolide has been isolated from the aerial part, leaves, and roots of *A. paniculata* with different solvents, including methanol, ethanol, and ethyl acetate [38,39].

### 3.5. 14-Deoxy-11,12-didehydroandrographolide

14-Deoxy-11,12-didehydroandroandrographolide is another major compound that is found in *A. paniculata*. It can be found in all parts of the plant but is predominantly found in the leaves. 14-Deoxy-11,12-didehydroandroandrographolide has a chemical formula of C_20_H_28_O_4_ and it forms a white crystal with a melting point of 204–205 °C. It showed UV spectra at 252 nm in a methanol solution [38,40,41].

## 4. Antiviral Activity

Viruses are small infectious agents that can only reproduce in living host cells; they are believed to have co-existed with living things since the beginning of life [40]. In the last two years, a new virus that infects the respiratory tract emerged, namely, the novel coronavirus SARS-CoV-2 (Severe acute respiratory syndrome coronavirus 2), which caused new diseases called COVID-19 and showed symptoms that was similar to SARS-CoV infection. The new virus has infected 232.72 million people and killed 4.76 million people worldwide as of 29 September 2021 [41].

The life journey of a virus that continues to mutate has caused viral infections to become resistant to several drugs and, thus, become the focus of the world’s health issues. Herbal plant-based medicine has made a significant contribution to developments in the medical sector, one of which is *A. paniculata*. The main component of this plant is andrographolide, which exhibits a broad range of antiviral activity. An evaluation of the potential of *A. paniculata* and its main components as antiviral agents will provide new insights into antiviral drug discovery. The antiviral activities *of A. paniculata* was summarized in Table 1, including their anti-dengue, anti-influenza, anti-HIV, anti-herpes simplex and anti SARS-CoV-2.

### 4.1. Anti-Dengue Virus

Dengue virus is a threat that continues to affect humans and has become endemic in several countries for decades. Dengue virus spreads through *Aedes aegypti* and *Aedes albopictus* mosquitoes [42]. Dengue virus causes self-limiting febrile illness, dengue hemorrhagic fever, and dengue shock syndrome [43]. Treatments for dengue disease are still limited. The search for drugs that are based on natural ingredients is a surefire method for fighting dengue disease.

One of the herbal plants that have been tested for its anti-dengue activity was *A. paniculata**,* with andrographolide as the main component. Andrographolide was reported to reduce cell infection and suppress dengue virus (DENV) production [44]. In its inhibitory phase, andrographolide suppresses viral production in the post-infection phase and reduces viral infections. At the stage of virus entry, andrographolide administration did not show significant inhibition on the HepG2 and Hela cell lines, meaning that andrographolide did not inhibit or prevent the dengue virus from infecting cells [45].

Furthermore, during its inhibition process, andrographolide upregulates heme oxygenase-1 (HO-1). HO-1 is an inducible enzyme in the heme catabolic pathway that can protect against the oxidative stress that is caused by the formation of reactive oxygen species (ROS) in DENV’s infection process. HO-1 induction can also inhibit protein synthesis and RNA replication in DENV serotypes 1–4. On the other hand, the overexpression of HO-1 degrades heme to form biliverdin, carbon monoxide (CO), and Fe^3+^, leading to the production of cytoprotective agents. However, in this case, only biliverdin acts as a cytoprotective agent against DENV because it inhibits DENV NS2B/NS3 protease activity, thus preventing DENV from replicating [46].

In the assay of andrographolide activity as an anti-DENV agent on several cell lines, approximately 15.62 µg/mL as a minimum non-toxic dose of andrographolide showed 97.23% inhibition against DENV2 on the C6/36 cell line [13]. Moreover, andrographolide reduced cell infection and virus production with 50% effective concentrations (EC_50_) on DENV2 of 21.304 µM and 22.739 µM for HepG2 and HeLa cell lines, respectively [45]. In addition, the methanolic extract from *A. paniculata* inhibited DENV in the Vero E6 cell line with an IC_50_ value of 20 µg/mL [47].

### 4.2. Anti-Influenza a Virus

Influenza A virus (IAV) attacks the respiratory system of humans and birds. IAV increases the population of cytokines that cause inflammation in the respiratory tract (especially in the alveolar compartment), acute respiratory syndrome, and even death [48,49]. Patients with IAV infection can transmit the virus through the air, as the viral replication process occurs in epithelial cells in the lower and upper respiratory tract [50].

Research on *A. paniculata* and its main component (andrographolide, which protects against influenza) was carried out. Andrographolide inhibits IAV during the gene replication and functionally mature protein phases. However, 14-deoxy-11,12-didehydroandrographolide, the second major component of *A. paniculata*, can inhibit the virus during the entry phase. In addition, both andrographolide and 14-deoxy-11,12-didehydroandrographolide reduced the expression of pro-inflammatory cytokines and chemokines that are caused by infection and eventually reduced the lung pathology caused by IAV (H1N1, H5N1, and H9N2) [3].

IAV has two surface glycoproteins: hemagglutinin (HA) and neuraminidase (NA). HA binds to sialic acid receptors on host cells, causing infection, while NA cleaves the receptor to be released from host cells. Molecular docking was carried out by Raja et al. [51], who showed that andrographolide binds to HA and NA by forming five or three hydrogen bonds, respectively (Figure 3). The binding indicates that andrographolide can interact with the proteins that are used by viruses during infection [51,52].

Meanwhile, via its inhibitory mechanism, 14-deoxy-11,12-didehydroandrographolide strongly inhibits H5N1 replication by reducing nucleoprotein (NP) mRNA, NP, and NS1 proteins. 14-Deoxy-11,12-didehydroandrographolide also effectively suppresses the nuclear export of viral ribonucleoprotein (vRNP) complexes during cell replication [53]. A decrease in the rate of cell replication causes a significant decrease in the expression of pro-inflammatory cytokines/chemokines.

In addition to inhibiting viral replication and production, andrographolide and 14-deoxy-11,12-didehydroandrographolide decrease inflammation by suppressing the proliferation of cytokines (TNF-α, IL-6, IL-8, IFN-α, IL-1β, and IFN-β) and chemokines (CXCL-10 and CCL-2) that are induced by IAV virus infection. They do this through several mechanisms, including inhibiting the NF- κ B signaling pathway, which plays a crucial role in cytokine expression. In other cases, both andrographolide and 14-deoxy-11,12-didehydroandrographolide downregulate the Janus kinase/signal transducer and transcription (JAK/STAT) activation signals that are stimulated by IAV. Within the JAK/STAT signaling pathway (induced by IFNs), hundreds of interferon-stimulated genes (ISGs) are upregulated [54]. This pathway is also a critical signaling pathway for inflammation in the lungs [55,56].

The results presented by Yu et al. [57] show that andrographolide exhibited 43.90 ± 2.49% viral inhibition activity by 250 µg/mL on a human bronchial epithelial cell line (16HBE) [57]. In addition, the IC_50_ values of 14-deoxy-11,12-didehydroandrographolide for reducing the cytopathic effect (CPE) that is caused by IAV infection were 5 ± 1 and 38 ± 1 µg/mL, in A549 and MDCK cells, respectively. On the other hand, 14-α-lipoyl andrographolide (AL-1), which is an andrographolide derivative, exerts robust anti-IAV activity with EC_50_ values of 8.4 µM (H9N2), 15.2 µM (H5N1), and 7.2 µM (H1N1) on the MDCK cell line [58].

### 4.3. Anti-HIV

HIV is a member of the Retroviridae family. HIV is spherical with a diameter of 120 nm. It attacks macrophage cells, such as monocytes, dendritic cells (DCs), and microglial cells. HIV infection in the immune system can lead to acquired immunodeficiency syndrome (AIDS), which causes a weakening of the immune system leads to various diseases that quickly attack the patient’s body. HIV can infect host cells when the glycoprotein gp120, which acts as the hand of HIV, binds to the CD4 receptor of the host cell. However, the infection does not occur quickly because the coreceptors on the surface of T cells, namely, C-X-C receptor 4 (CXCR4) and C-C receptor 5 (CCR5), must take part in the process (Figure 4) [59].

Sixteen Chinese herbal medicines, including andrographolide, were analyzed in vitro and in vivo for their ability to downregulate CXCR4 and CCR5 coreceptors in T cells. A 70% ethanolic extract of *A. paniculata* significantly downregulated the CXCR4 and CCR5 promoters with an EC_50_ value of 5.49 µg/mL. *A. paniculata* can also suppress cell fusion and decrease p24 antigen (p24 antigen levels were correlated with the proportion of immunological failure). Clinical trials using human T cells showed a decrease in CXCR4 and CCR5 levels from 35 to 10% and 25 to 10%, respectively, via flow cytometry after administering *A. paniculata* extract [60].

In the same year, the main component of *A. paniculata*, namely, andrographolide, was found to inhibit gp120-mediated cell fusion in HL2/3 cells with an IC_50_ value of 0.59 M. Molecular modeling results show that andrographolide binds to the V3 loop of gp120. The andrographolide binding indicates that it could be an HIV infection prevention agent [61]. In addition, andrographolide and 14-deoxy-11,12-didehydroandrographolide showed good anti-HIV activity with EC_50_ values of 49.0 and 56.8 µg/mL, respectively [62]. 

### 4.4. Anti-Herpes Simplex

The herpes virus belongs to the *Herpesviridae* family, with herpes simplex being the most common virus found in humans. Herpes simplex virus (HSV) attacks the skin and mucosal epithelial cells. People infected with HSV get blisters on the skin and mucous membranes in the mouth, lips, genitals, and throat. In the eye area, HSV infection causes conjunctivitis, keratitis, iridocyclitis, and acute retinal necrosis. HSV infection is usually treated with acyclovir (ACV), but recently, HSV has become resistant to ACV [21,63]. New drug candidates are needed for treating HSV to overcome the mutated virus. 

Andrographolide and its derivatives exhibit anti-HSV activity. Andrographolide, neoandrographolide, and 14-deoxy-11,12-didehydroandrographolide showed virucidal activity in HSV-1 without showing significant cytotoxic effects at virucidal concentrations (IC_50_ values of 8.28, 7.97, and 11.1 µg/mL, respectively) [64]. In addition, andrographolide and 14-deoxyandrographolide inhibit the viral entry stage and suppress viral production during the replication stage by interfering with early gene expression through glycoproteins C and D, which are produced by the early gene [65].

Another type of herpes virus is the Epstein–Barr virus (EBV). In the early stages of its lytic cycle, EBV requires BRLF1 and BZLF1, which are immediate–early genes that encode transcription factors Zta and Rta. Both of these transcription factors have important roles in shaping lytic gene sets, thereby affecting the encoding of antigen (EA-D) and DNA polymerase [66]. Therefore, inhibiting the formation of transcription of these factors can suppress the rate of virus production because the reproductive cycle is disrupted.

An ethanol extract of *A. paniculata* (EEAP) and andrographolide (25 and 5 µg/mL, respectively) significantly inhibited the expression of EBV lytic proteins, Rta, Zta, and EA-D during the viral lytic cycle in P3HR1 cells [67]. Thus, the mechanism of inhibition of the EEAP and andrographolide against EBV was to inhibit BRLF1 and BZLF1, thereby stopping the lytic cycle of EBV (Figure 5).

### 4.5. Anti-SARS-CoV-2

SARS-CoV-2 has caused a pandemic that includes almost all countries around the world. It attacks the respiratory organs and spreads very quickly. The search for drugs is still ongoing because this virus is likely to continue to co-exist with humans. Several methods have been used to search for SARS-CoV-2 drugs, including in silico (computational) approaches and in vitro studies [20].

One way to inhibit SARS-CoV-2 is to inhibit the main protease (M^pro^), 3C-like proteinases (3CL^pro^), and papain-like protease (PL^pro^). These non-structural proteins are viral proteases that are crucial to the production of functional polyproteins, which are needed for viral RNA replication and transcription. In addition, SARS-CoV-2 requires cellular receptor angiotensin-converting enzyme 2 (ACE2) to enter the host cell. By inhibiting the formation of non-structural proteins from SARS-CoV-2, the replication or production of the virus can be suppressed. In addition, blocking the ACE2 receptor on the host cell prevents the virus from penetrating the cell [68,69,70,71].

So far, in silico research has identified the main enzyme of SARS-CoV-2, a spiked glycoprotein, and a cell receptor. The spike glycoprotein in SARS-CoV acts as a viral antigen that is responsible for binding to host receptors, internalizing the virus, and inducing strong humoral and cellular immune responses in humans during infection [70,72]. The ACE 2 receptor and spiked glycoprotein have Moldock scores of 99.354 and 98.80 kcal/mol, respectively [71]. Meanwhile, molecular docking with the Glide module showed that andrographolide has a binding affinity for the main protease SARS-CoV-2, with a Glide score of 6.26 [73]. In addition, four main components of *A. paniculata*, namely andrographolide, 14-deoxy-11,12-didehydroandrographolide, neoandrographolide, and 14-deoxyandrographolide, have the potential to protect against the 3-chymotrypsin-like protease (3CL^pro^), spike, RNA-dependent RNA polymerase (RdRp), and papain-like protease (PL^pro^) of the virus, which is responsible for replication, transcription, and host cell recognition [74,75].

Recently, in vitro assays of Calu-3 cells infected with SARS-CoV-2 have shown that *A. paniculata* and andrographolide significantly inhibited the production of infectious virions with an IC_50_ of 0.036 µg/mL and 0.034 M, respectively [68]. Subsequently, andrographolide suppressed the main protease (M^pro^) activity of SARS-CoV-2 and SARS-CoV via a cleavage assay, yielding IC_50_ values of 15.05 ± 1.58 and 5.00 ± 0.67 µM, respectively [69].

## 5. Delivery System

The use of andrographolide as a therapeutic agent, especially as an antiviral agent, is still limited. However, in vitro andrographolide has potential as an agent for treating various types of diseases [3,9,11,12]. Furthermore, this potential is limited to in vitro tests. Its application to living things will be much different considering the shortcomings of andrographolide. Andrographolide has an α,β-unsaturated γ-lactone ring connected to the decalin ring through unsaturated C_2_ (Figure 2) [26]. It impacts the solubility of water and has poor oral absorption and low bioavailability because it is not stable under the acidic or alkaline conditions of the digestive system [26]. The low bioavailability of andrographolide significantly interferes with the performance of andrographolide to treat diseases. The absorption process of andrographolide will also be disrupted such that the concentration of andrographolide in the blood is insufficient to achieve optimal treatment activity. 

The primary purpose of using a delivery system is to increase the bioavailability of andrographolide and provide a time-release effect on the absorption process to extend the half-life of andrographolide in the blood [76]. The application of drugs that do not use a carrier yields a short half-life since the body absorbs the drug component according to its ability when it enters the body. Generally, the drug compound that is not absorbed will be excreted through the excretory channel [77]. Therefore, several researchers have modified the use of andrographolide in the delivery system. The types of drug delivery system of andrographolide reviewed in this study are summarized in Table 2, including microsphere, microemulsion, liposome, noisome, and nanoparticles, as well as their biocompatibility aspects.

### 5.1. Microsphere

A microsphere is a spherical microparticle with sizes ranging from 1–1000 µm. Microscale particles are widely used as adsorbents because of their high surface area and high surface-to-volume ratio. Based on its ability as an adsorbent, the microsphere can absorb drug compounds and be used as a drug carrier [78,79,80]. Microspheres formulated with polylactic co-glycolic acid (PLGA) can be used as a delivery medium for andrographolide.

PLGA is a biodegradable polymer that is frequently used for developing microspheres. Andrographolide-loaded microspheres formulated with PLGA and prepared using the solvent emulsion evaporation method can provide a sustained time-release effect on andrographolide, increasing its bioavailability.

In-vitro test results show that an andrographolide-loaded PLGA microsphere with a diameter of 53.18 ± 2.11 µm can sustain release for up to nine days. An andrographolide-loaded microsphere achieves a maximum plasma concentration (C_max_) of 28.44 ± 3.76 ng/mL at a maximum time (t_max_) of 23.73 ± 2.25 h, presenting a half-life (t_1/2_) of 24.84 ± 3.01 h. In contrast, pure andrographolide reaches its highest concentration at the beginning of administration (less than an hour) and drops immediately after the next hour. Its half-life is only 0.054 ± 0.008 h [81]. The release time of andrographolide can be long due to its dissolution mechanism through the polymer’s diffusion and the polymer layer’s erosion on the microsphere [82]. 

However, an andrographolide-loaded microsphere still has a few drawbacks. For instance, removing the organic solvent through microsphere preparation is difficult because it can lead to toxicity in normal cells if not carried out thoroughly.

### 5.2. Microemulsion 

A microemulsion is a dispersion of oil, water, and surfactant that forms an isotropic and thermodynamically stable system. Microemulsions have domain diameters ranging from 10–100 nm [83]. The microemulsion method can increase the bioavailability of oral drugs because it combines oil, water, and surfactant systems (Figure 6). The solubility of the drug compound increases because it can dissolve in two fused systems: water (polar) and oil (non-polar) [84]. This solubility in two fused systems is necessary to overcome the low bioavailability of andrographolide when it is administered orally.

The process of loading andrographolide into microemulsions during its application was studied by several researchers. Du et al. [85] used alcohol (co-surfactant), Tween 80 (surfactant), isopropyl myristate (oil), and aqua bides (water) as microemulsion agents. In addition, Sermkaew et al. [86] used capryol, cremphor, and labrasol, while Syukri et al. [87] used capryol, Tween 20, and polyethene glycol (PEG) 400 as microemulsion agents. The particle sizes range from 10–25 nm. The solubility of andrographolide in the microemulsion increased significantly and it was stabilized across time, temperature, and different gravity states while its bioavailability increased.

An in vivo study by Syukri et al. [87] showed that using an andrographolide-loaded microemulsion enhances andrographolide’s bioavailability compared to free andrographolide. The areas under the curve (AUCs) of free andrographolide and andrographolide-loaded microemulsion were 3.72 ± 0.3 and 4.70 ± 0.19, respectively, and the C_max_ values were 1.53 ± 0.14 and 1.99 ± 0.20 µg/mL, respectively [87]. On the other hand, the lethal dose (LD_50_) of andrographolide-microemulsion was found to be 138.36 mg/kg, which is 266 times the daily oral dose for adult Kunming mice (0.52 mg/kg). With the increase in solubility and bioavailability, the drug dose will be reduced, which means that the toxic effects of oral drug use can be eliminated [85].

### 5.3. Liposomes

Another method that is used to increase the effectiveness with which drugs penetrate cell membranes and their solubility is to use liposomes as the drug delivery system. Liposomes are artificial vesicles that consist of a lipid bilayer membrane (Figure 7). Liposomes can be prepared with phosphatidylethanolamine (PDEA), cholesterol, dicetyl phosphate (DCP), and andrographolide with a molar ratio of 7:1:1:1 using chloroform and methanol (2:1 *v/v*) as solvents via the thin-film hydration method [88]. In addition, liposomes can be prepared with andrographolide using the same method with soybean phosphatidylcholine (SPC), cholesterol, and DSPE-PEG2000-Mal with a molar ratio of 25:11:0.7 and dissolved in chloroform.

After that, andrographolide was added and dissolved in ethanol at a ratio of 1:20 (*w/w*) [89]. Andrographolide encapsulated with the liposomal system showed increased accumulation in tumor tissue and deep intratumoral penetration rates [90]. Andrographolide-loaded liposomes showed a higher cytotoxicity effect than free andrographolide. The apoptosis rates of the lung cancer cells were 39.9, 45.3, and 69% for the free combo drugs (doxorubicin and andrographolide), DSPE-PEG2000-modified liposome (Lipo-PEG), and cell-penetrating peptides-modified liposome (Lipo-CPP), respectively [89].

### 5.4. Niosomes

Niosomes provide a relatively new method that is used as the drug delivery system. The purpose of using niosomes to encapsulate drug compounds is to increase drug bioavailability and improve tissue distribution [91]. Niosomes are vesicles that are formed from non-ionic surfactants consisting of a hydrophilic head and a hydrophobic tail (Figure 7). Loading andrographolide into a niosome can improve its bioavailability and tissue distribution. Andrographolide-loaded niosomes were prepared via a film hydration/sonication method using Span 60 (50 mg), cholesterol (7.35 mg), and andrographolide (5 mg) [92]. They were then hydrated with 20% propylene glycol and sonicated to reduce the particle size [92,93] to 206 nm, which can be used for targeting the liver (drugs for targeting the liver must be approximately 200 nm in size) [94].

The obtained andrographolide niosome has an encapsulation efficacy of 72.36% and a drug loading ratio of 5.90%. The encapsulation of andrographolide into the niosome can reach most of the body tissues analyzed by HPLC. Niosome facilitates the distribution of andrographolide throughout the tissue. In addition, the niosome facilitates cell penetration because it has a bilayer membrane with the same properties as the cell membrane (Figure 7). Therefore, andrographolide is suitable for its intended use [92].

On the other hand, the toxicity of the drugs was reduced because they were encapsulated [92]. However, the drawback of this method is related to the low drug loading ratio, meaning that it still requires high doses. In addition, both free andrographolide and andrographolide-loaded niosomes had no significant difference in their activity against hepatocellular carcinoma (HCC). The blank niosome had no toxicity effect, but the IC_50_ values of free andrographolide and andrographolide-loaded niosomes are 25.2 and 25.0 µM, respectively [92]. Compared to the liposome, the niosome did not significantly increase the activity of andrographolide, whereas the liposome increased the activity of andrographolide and doxorubicin compared to the free andrographolide. However, niosome application can increase the bioavailability of andrographolide in biological systems

### 5.5. Nanoparticles

The drawback of the niosome system, namely, the low drug loading ratio, can be overcome by using nanoparticles for drug delivery [95,96]. The synthesis of nanoparticles as an andrographolide delivery system was carried out using solid lipid nanoparticles (SLN) with Compritol 888 ATO as the solid lipid and Brij 78 as the surfactant, followed by emulsification, evaporation, and solidification processes [97,98]. In addition, nanoparticles made from polymers were also carried out for the andrographolide delivery system, in this case, PLGA (poly(lactic-co-glycolic) acid), which was prepared using the single emulsion evaporation method (Figure 8) [99,100].

SLN has a high encapsulation efficiency (92%) and a 262–278 nm particle size. The use of nanoparticles can also expand tissue distribution, which reaches the blood–brain barrier [97]. Andrographolide-loaded nanoparticles exhibit high physical and chemical stability. Storage within a month in lyophilized powder or an aqueous dispersion at 4 and 25 °C showed no significant changes in particle size, polydispersity index (PDI), or zeta potential value.

In addition, andrographolide that is embedded in nanoparticles can produce a slow-release effect because the dissolution system refers to the diffusion and dissolution of andrographolide from the nanoparticle matrix. Andrographolide-loaded nanoparticles showed a slower release rate than the solution form of andrographolide but a faster release rate than the suspension form [97,101].

## 6. Conclusions

*Andrographis paniculata* has been widely used throughout the world as an antiviral drug. Its mechanisms of action in inhibiting viral infection can be categorized into several categories, including regulating the viral entry stage, viral gene replication, and the formation of mature functional proteins. In addition, andrographolide suppresses the induction of cytokines and chemokines to prevent inflammation that can damage cells. To achieve entry into the host cell and have a pharmacological effect, andrographolide must have a high bioavailability and be thermodynamically stable. However, andrographolide has several shortcomings, including low bioavailability, solubility, and stability. Drug delivery development is the most effective way to overcome the unwanted pharmacological effects of bioactive substances from natural products, including their low solubility and stability. Suitable carriers that protect drugs from rapid degradation are essential components of drug delivery systems. Drug delivery systems are also designed to provide prolonged drug distribution with effective and precise kinetics over extended periods.

This review confirms that incorporating andrographolide into various delivery systems has unique characteristics, benefits, and drawbacks. In general, they are stable, exhibit increased activity, and prolong the drug’s lifetime. The use of microspheres and microemulsions as andrographolide drug delivery carriers is suitable for oral drug administration. The advantages of this delivery approach include improved absorption after oral administration.

Liposome and niosome vesicles are identified in this review as versatile carriers with numerous advantages, including easy encapsulation for hydrophilic and hydrophobic drugs, increased bioavailability, and extended drug lifetimes. In addition, the use of niosomes can improve the bioavailability of andrographolide in biological systems compared to liposomes. Andrographolide-loaded nanoparticles provided a high drug loading capacity and showed higher stability after long storage times. Using various types of drug delivery systems for natural compounds will expand the range of herbal drug developments.

## 7. Future Prospects

Delivery systems based on andrographolide (and its derivatives) have not been studied comprehensively, even though it exhibits a wide range of pharmacological effects, including antiviral, anticancer, anti-diabetic, anti-hypertension, and wound-healing effects. Delivery systems of andrographolide and its derivatives is an interesting research topic because this natural product has the same shortcomings as other natural products, including its low bioavailability and stability. Thus, studies on delivery systems of andrographolide and its derivatives are urgently needed to overcome the shortcomings of the application of this natural compound.

## Figures and Tables

**Figure 1 pharmaceuticals-14-01102-f001:**
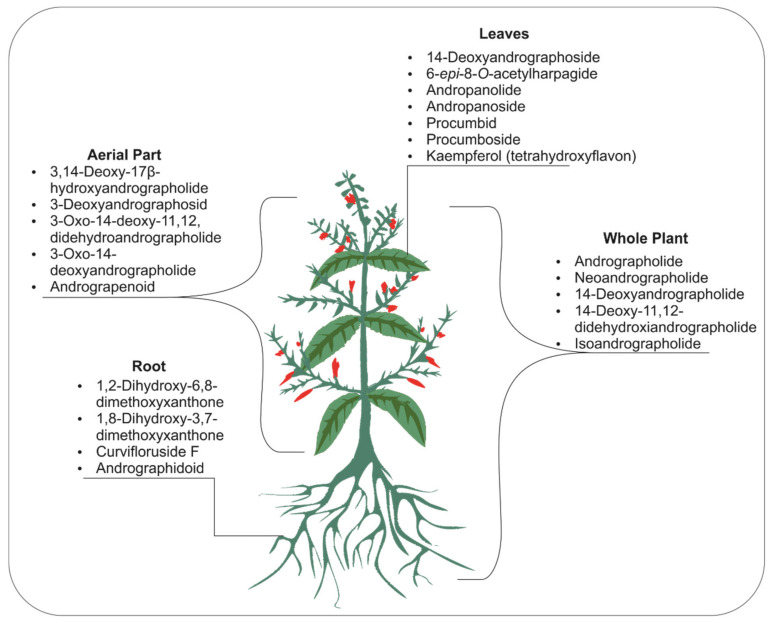
The compounds contained in *A. paniculata*, divided into four parts: whole plant, aerial plant, leaves, and roots. The whole plant comprises all parts of the plant. The aerial part is that which is directly exposed to the atmosphere, including stems, twigs, leaves, and flowers.

**Figure 2 pharmaceuticals-14-01102-f002:**
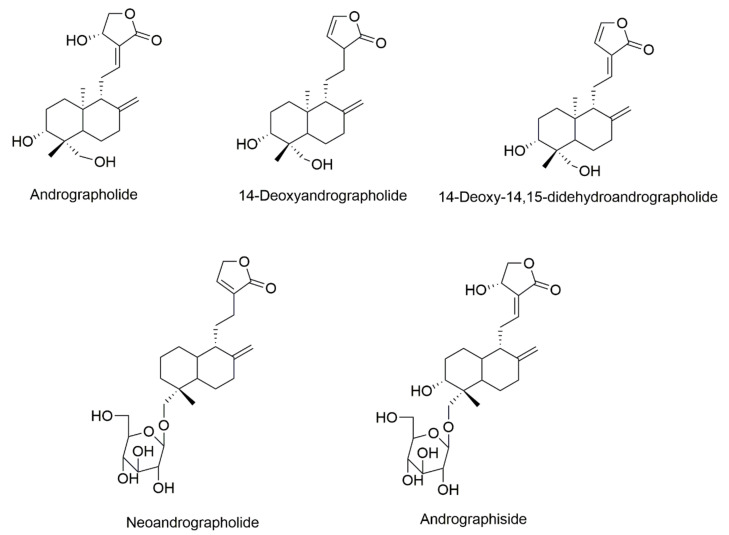
Chemical structures of the main components of *A. paniculata*.

**Figure 3 pharmaceuticals-14-01102-f003:**
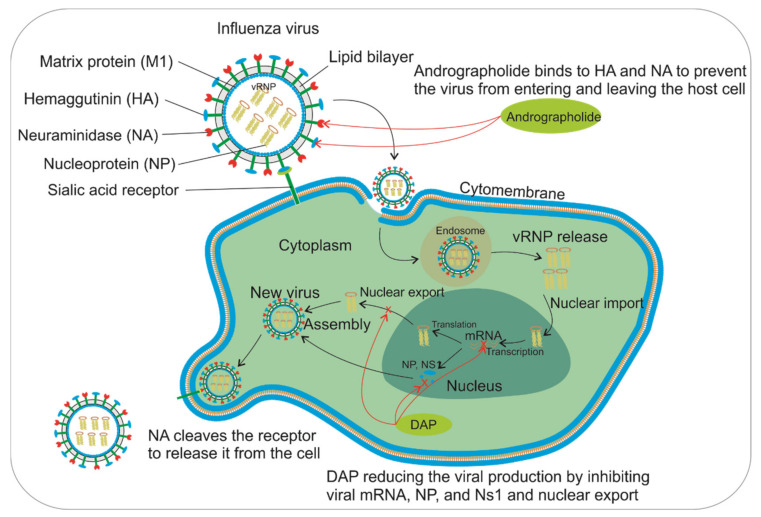
The processes by which IAV invades the host cell and andrographolide and 14-deoxy-11,12-didehydroandrographolide (DAP) inhibit IAV replication and prevent IAV invasion. Andrographolide treatment inhibits viral invasion by binding with HA and NA so that the sialic acid receptor cannot bind to HA or NA. 14-Deoxy-11,12-didehydroandrographolide (DAP) treatment inhibits viral production by suppressing the NP, NP1, mRNA, and nuclear export through the inner cells. Andrographolide and DAP need to be distributed and penetrate the cell to inhibit viral invasions and production after oral administration.

**Figure 4 pharmaceuticals-14-01102-f004:**
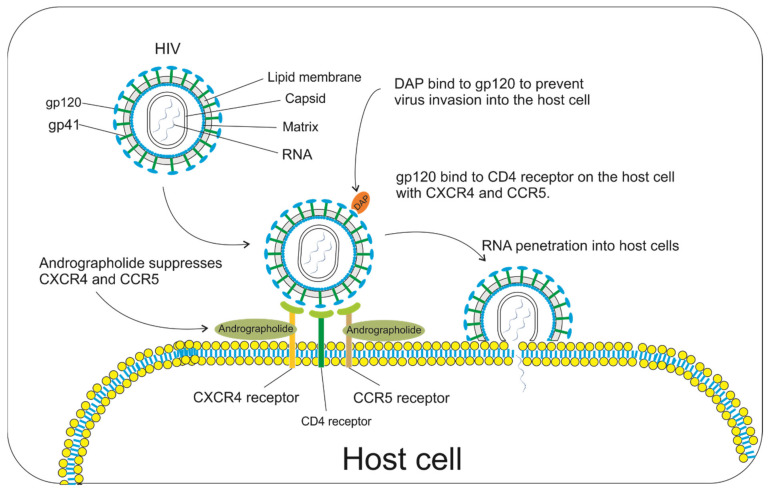
The process of HIV invasion into the host cell and andrographolide’s inhibition of virus infection. Andrographolide suppresses the CXCR4 and CCR5 receptors to prevent HIV from entering the cell. Meanwhile, 14-deoxy-11,12-didehydroandrographolide (DAP) binds to gp120 in HIV to reduce the virus’s interaction with the cell receptor (CD4) because the active site of the viruses is blocked by DAP [3].

**Figure 5 pharmaceuticals-14-01102-f005:**
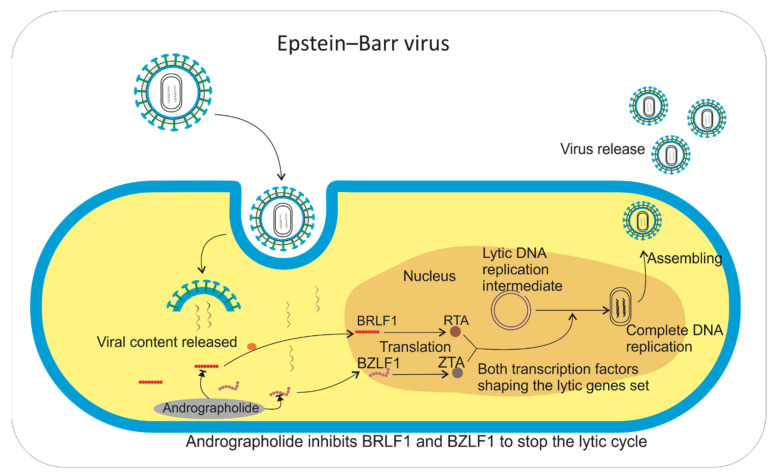
The process that takes place in host cells that are infected by EBV. Andrographolides inhibit BRLF1 and BZLF1, which are required to conduct transcription factors RTA and ZTA. RTA and ZTA are important for completing DNA replication.

**Figure 6 pharmaceuticals-14-01102-f006:**
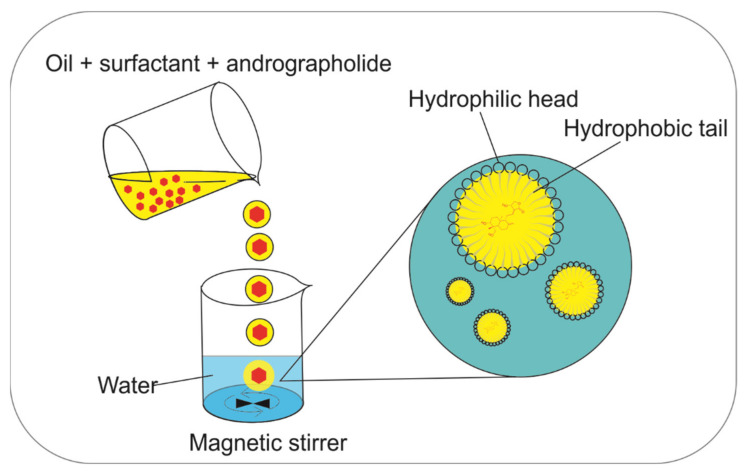
The preparation process that is used to create an andrographolide-loaded microemulsion.

**Figure 7 pharmaceuticals-14-01102-f007:**
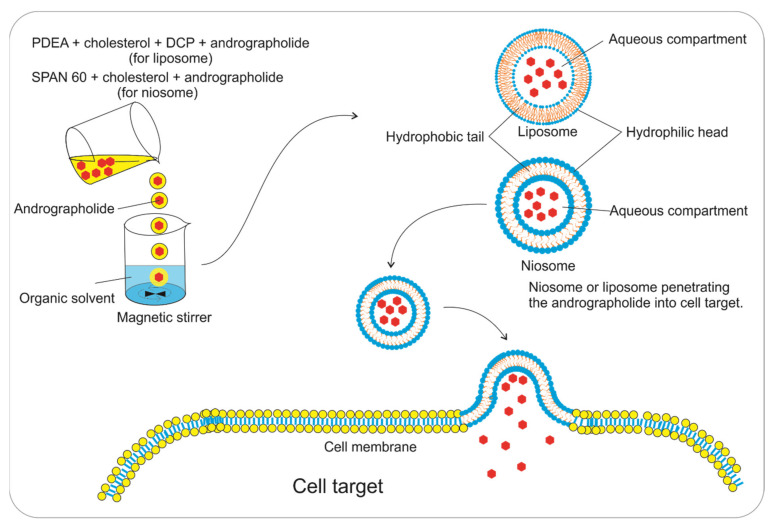
The preparation and delivery of andrographolide-loaded liposomes and niosomes.

**Figure 8 pharmaceuticals-14-01102-f008:**
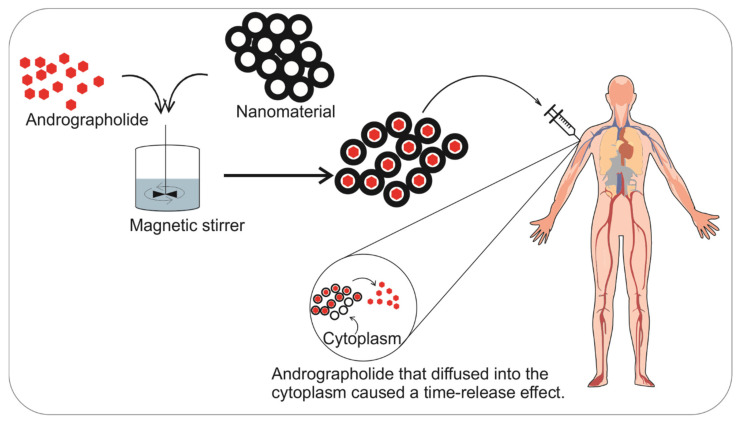
The creation and activation process of andrographolide-loaded nanoparticles. Nanoparticles produce a time-release effect of andrographolide via passive diffusion between andrographolide and the cytoplasm.

**Table 1 pharmaceuticals-14-01102-t001:** Antiviral activities of chemical compound and extract from *A. paniculata*.

Antivirals	*A. paniculata* (Extract and Chemical Compounds)	Cell Target	Inhibition Activity	Ref.
Anti-dengue	Andrographolide	C6/C3 cell line	97.23% viral inhibition using 15.62 µg/mL	[13]
	Andrographolide	HepG2 and HeLa cells	Reduce cell infection and viral production with EC_50_ values of 21.304 and 22.739 µM, respectively	[45]
	Methanolic extract	Vero E6	Inhibits DENV with an IC_50_ value of 20 µg/mL	[47]
Anti-influenza	Andrographolide	Human bronchial epithelial cell line (16HBE)	43.90 ± 2.49% viral inhibition by 250 µg/mL	[57]
	14-Deoxy-11,12-didehydroandrographolide	A549 and MDCK cells	Reduce cytopathic effect (CPE) with IC_50_ values of 5 ± 1 and 38 ± 1 µg/mL, respectively	[58]
Anti-HIV	Ethanolic extract	Human T cell	Downregulate CXCR4 and CCR5 with an EC_50_ value of 5.49 µg/mL	[60]
	Andrographolide	HL2/3 cell	Inhibits gp120-mediated cell fusion with an IC_50_ value of 0.59 M	[61]
		MT2 cell	Inhibits the p24 antigen with an EC_50_ value of 49.0 µg/mL	[62]
	14-Deoxy-11,12-didehydroandrographolide	MT2 cell	Inhibits the p24 antigen with an EC_50_ value of 56.8 µg/mL	[62]
Anti-herpes simplex	Andrographolide	Vero cell	Inhibits the cytocidal effect with an IC_50_ value of 8.28 µg/mL	[64]
	Neoandrographolide	Vero cell	Inhibits the cytocidal effect with an IC_50_ value of 7.97µg/mL	
	14-Deoxy-11,12-didehydroandrographolide	Vero cell	Inhibits the cytocidal effect with an IC_50_ value of 11.1 µg/mL	
Anti-SARS-CoV-2	Ethanolic extract	Calu-3 cell	Inhibits viral production with an IC_50_ value of 0.036 µg/mL	[68]
	Andrographolide		Inhibits viral production and suppresses the main protease (M^pro^) activity with IC_50_ values of 0.034 and 15.05 ± 1.58 µM, respectively.	[68,69]

**Table 2 pharmaceuticals-14-01102-t002:** The andrographolide delivery system.

Type of Drug Delivery	Formulation	Method	Biocompatibility Aspects	Ref
Microsphere	PLGA (polylactic co-glycolic acid) and andrographolide	Emulsion solvent evaporation	Prolonged release (up to nine days) Increases the half-life of andrographolide	[84,85]
Microemulsion	Alcohol, Tween 80, isopropyl myristate, water, and andrographolide	Spheronization technique	Increases the solubility Stabilized over time, temperatures, and different gravity states Low acute oral toxicity	[85]
	Capryol, cremphor, labrasol, and *A. paniculata* extract	Extrusion/spheronization technique	Slow release of andrographolide Increases the oral absorption	[86]
	Capryol, Tween 20, PEG (polyethene glycol) 400, and andrographolide	Spheronization technique	Increases the stability, and improves the andrographolide bioavailability	[87]
Liposome	Phosphatidylethanolamine (PDEA), cholesterol, and dicetyl phosphate (DCP)	Thin-film hydration method	Higher cytotoxic effect Increases the accumulation in tumor tissue	[88]
	Soybean phosphatidylcholine (SPC), cholesterol, and DSPE-PEG2000-Mal		Increases the solubility of andrographolide	[89]
Niosome	Span 60 (50 mg), cholesterol (7.35 mg), and andrographolide (5 mg)	Film hydration/sonication method	Increases the andrographolide absorption Reduce toxicity	[92,93]
Nanoparticles	Compritol 888 ATO, Brij 78, and andrographolide	Emulsion/evaporation/solidifying	Expands the tissue distribution Excellent physical and chemical stability during storage Slow-release effect	[97,98]
	PLGA (poly(lactic-co-glycolic) acid) and andrographolide	Emulsion evaporation	Slow-release effect	[99,100]

## Data Availability

Not applicable.

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
