# Peer review of "Antiviral Activities of Andrographolide and Its Derivatives: Mechanism of Action and Delivery System"

_pharmaceuticals, 2021, doi:10.3390/ph14111102_

Round 1

Reviewer 1 Report

The article of Sya'ban Putra Adiguna et. al. entitled “Antiviral Activities of Andrographolide and Its derivatives: Mechanism of Action and Delivery System” describes an informative study that aims to highlight the phytoconstituents, physicochemical characteristics, and delivery systems. This is a comprehensive review explained about antiviral mechanisms of Andrographolide and Its derivatives, however, some concerns should be addressed by the Authors

  1. The authors should explain the antiviral mechanism of andrographolide and its analogs on SARS-CoV-2 as they mentioned in the abstract.
  2. Authors can make separate tables of in vivo and clinical studies on antiviral activities of A. paniculata.
  3. Authors can provide the sources of references or data obtained for review writing.
  4. Autor must find appropriate references and eradicate ‘Error! Reference source not found’ in the text
  5. Toxicity studies, biocompatibility, biodegradability, stability, and half-life in the portal circulation, renal clearance, accumulation, and uptake of Andrographolide and Its derivatives have to be included in the text
  6. The authors should provide their justification and relevance of the study. This will help the readers to understand the importance of the paper. Relevant articles in the field such as Life 2021, 11, 348;  J. Ethnopharmacol. 2021, 272, 113954; J. Ethnopharmacol. 2021, 275, 114054; . Antiviral Res. 2017,139, 69–78; Biol Pharm Bull.2009 Aug;32(8):1385-91, HAYATI Journal of Biosciences, 22(2), 2015, 67-72.; Asian Pac J Trop Dis 4 (2), 2014, S624-S630, may be discussed to improve the review.
  7. Significant grammar and typographical errors were found throughout the manuscript and should be corrected. For instance: Fig. 6: hydrophobic tail, magnetic stirrer

Author Response

Responses to Reviewers` comments

Manuscript ID Number: pharmaceuticals-1428332

Manuscript Title: Antiviral Activities of Andrographolide and Its derivatives: Mechanism of Action and Delivery System

Below are our responses to the comments of reviewers. We have copied and pasted their comments. The reviewer’s comments appear in italic. Our responses appear in regular font.

Responses to Reviewer 1

Comment:

The authors should explain the antiviral mechanism of andrographolide and its analogs on SARS-CoV-2 as they mentioned in the abstract.

Response:

Thank you for your remarks on mechanism of action related to the anti-SARS-Cov-2 activities. We have added an explanation of the mechanism of action of andrographolide in inhibiting SARS-CoV-2 replication and cell penetration

Comment:

Authors can make separate tables of in vivo and clinical studies on antiviral activities of A. paniculata.

Response:

Authors can provide the sources of references or data obtained for review writing.

Response:

Thank you for your remarks. We have added the search strategy and marked as a second section.

Comment:

Autor must find appropriate references and eradicate ‘Error! Reference source not found’ in the text

Response:

Thank you for your remarks. We have included several sources from the book section, which are references number 34 and 85. We have also cross-checked the references mentioned in lines 51, 102, 128 where there are numbers in brackets that may be considered as citations. this number represents the number of compounds that have been found in A. paniculata. But after discussion, we decided to remove the number to avoid reference error.

Comment:

Toxicity studies, biocompatibility, biodegradability, stability, and half-life in the portal circulation, renal clearance, accumulation, and uptake of Andrographolide and Its derivatives have to be included in the text

Response:

Thank you for your remarks. The requested data is not all available in every section, so we write only available data such as half-life time, stability, and biocompatibility.

Comment:

The authors should provide their justification and relevance of the study. This will help the readers to understand the importance of the paper. Relevant articles in the field such as Life 2021, 11, 348;  J. Ethnopharmacol. 2021, 272, 113954; J. Ethnopharmacol. 2021, 275, 114054; . Antiviral Res. 2017,139, 69–78; Biol Pharm Bull.2009 Aug;32(8):1385-91, HAYATI Journal of Biosciences, 22(2), 2015, 67-72.; Asian Pac J Trop Dis 4 (2), 2014, S624-S630, may be discussed to improve the review.

Response:

Thank you for your remarks and we already recheck the reference and most of the references that mentioned has been discussed in the text and we will make sure that the additional references will be added to our reference list.

Comment:

Significant grammar and typographical errors were found throughout the manuscript and should be corrected. For instance: Fig. 6: hydrophobic tail, magnetic stirrer

Response:

Thank you for your remarks. We already resolve the problem regarding the grammar and typographical error.

Reviewer 2 Report

Title: Antiviral Activities of Andrographolide and Its derivatives: Mechanism of Action and Delivery System

Manuscript ID: pharmaceuticals-1428332

Manuscript Type: Review Article

I am thankful to the journal for providing me the opportunity to review the article.

  1. paniculata has been widely used throughout the world, one of which is as an anti- viral drug. In this review, authors summarized that basically, the mechanism of action of andrographolide in inhibiting viral infection can be categorized into several mechanisms, including regulating the viral entry stage, viral gene replication and the formation of mature functional proteins

Review is written in very scientific and impressive way.

Here are my few concerns to the author regarding manuscript-

  1. I would like to suggest to authors for explaining the Figure 3. Authors explain that andrographolide attaches to the virus spikes and DAP is present in the cells. Here my concern is regarding mechanism and application of andrographolide. Would it be used like topical solution or what? The virus will be inhibited directly without enetering to the host cells via attachment of andrographolide directly to the spikes of virus. Is there any specific kind of competition between the target cell receptors and andrographolide. Kindly explain the figure appropriately.
  2. Give full form of DAP in text.
  3. The necessity and innovation of the article should be presented to the introduction.
  4. It is suggested to rewrite conclusion part in more crisp way. This section should present in one 250-300 words paragraph.
  5. Authors should also need to proceed for dynamical study in future path.
  6. There are lot of punctuation and typographical errors throughout in the manuscript. It must be rechecked by native English speaker.
  7. Author must be provide good high resolution/quality of picture.
  1. Match all the cited references in the text part.

I strongly recommend the review article for publications in the reputed journal after minor revision.

Author Response

Responses to Reviewers` comments

Manuscript ID Number: pharmaceuticals-1428332

Manuscript Title: Antiviral Activities of Andrographolide and Its derivatives: Mechanism of Action and Delivery System

Below are our responses to the comments of reviewers. We have copied and pasted their comments. The reviewer’s comments appear in italic. Our responses appear in regular font.

Responses to Reviewer 2

Comment:

I would like to suggest to authors for explaining the Figure 3. Authors explain that andrographolide attaches to the virus spikes and DAP is present in the cells. Here my concern is regarding mechanism and application of andrographolide. Would it be used like topical solution or what? The virus will be inhibited directly without enetering to the host cells via attachment of andrographolide directly to the spikes of virus. Is there any specific kind of competition between the target cell receptors and andrographolide. Kindly explain the figure appropriately.

Response:

Thank you for your remarks. We already explain the figure 3 clearly related to the application of andrographolide in the figure caption.

Comment:

Give full form of DAP in text.

Response:

Thank you for your remarks. we have added the full form of DAP in the figure caption.

Comment:

The necessity and innovation of the article should be presented to the introduction

Response:

Thank you for your remarks, we have added an explanation regarding the necessity and innovation in introduction.

Comment:

It is suggested to rewrite conclusion part in more crisp way. This section should present in one 250-300 words paragraph.

Response:

Thank you for your remarks. The conclusion has been rewrite according to the condition provided.

Comment:

Authors should also need to proceed for dynamical study in future path.

Response:

Thank you for your remarks. We have added one section after the conclusion related to future prospect.

Comment:

There are lot of punctuation and typographical errors throughout in the manuscript. It must be rechecked by native English speaker.

Response:

Thank you for your remarks. We will recheck the grammar and typographical error throughout the manuscript.

Comment:

Author must be provide good high resolution/quality of picture.

Response:

Thank you for your remarks. All illustration images are made with the Corel Draw X7 series application and have been exported at 100% scale into .tiff format before being inserted into the manuscript.

Comment:

Match all the cited references in the text part

Response:

Thank you for your remarks, We have cross-checked the references mentioned in lines 51, 102, 128 where there are numbers in brackets that may be considered as citations. this number represents the number of compounds that have been found in A. paniculata. But after discussion, we decided to remove the number to avoid reference error.

Round 2

Reviewer 1 Report

The article is now suitable for publication in Pharmaceuticals